# Did the severity of appendicitis increase during the COVID-19 pandemic?

Yao-Jen Chang[1,2], Li-Ju Chen[3,4], Yun-Jau Chang [5,6]*

1 Department of Surgery, Taipei Tzu Chi Hospital, Buddhist Tzu Chi Medical Foundation, New Taipei City, Taiwan, 2 School of Medicine, Buddhist Tzu Chi University, Hualien, Taiwan, 3 University of Taipei, Taipei, Taiwan, 4 Department of Ophthalmology, HepingFuyou Branch, Taipei City Hospital, Taipei, Taiwan, 5 Department of General Surgery, National Taiwan University Hospital, Taipei, Taiwan, 6 Department of General Surgery, Zhong-Xing Branch, Taipei City Hospital, Taipei, Taiwan

* yunjauchang2003@yahoo.com.tw

## Abstract

### Background

This study aimed to assess the severity of appendicitis during the coronavirus disease 2019 (COVID-19) pandemic, as patients with appendicitis may procrastinate seeking medical attention during the pandemic.

### Methods

Information on patients with appendicitis who were treated at the Taipei City Hospital during the COVID-19 pandemic (January 1, 2020 to June 30, 2020) was retrieved. Patients who were diagnosed with appendicitis and treated at the same hospital from January 1, 2019 to July 1, 2019 were designated as the control group. Multivariate logistic regression analysis was conducted to assess changes in the severity of appendicitis (at a 2-week interval) between the two groups.

### Results

We identified 307 (study group: 149; control group: 158) consecutive patients with appendicitis. The mean age was 46.2 +− 19.8 years. Between the two groups, there were no significant differences in age, sex, comorbidity, surgery type (laparoscopic or open appendectomy) or surgery time. The number of patients in the study group decreased between January 29, 2020 and April 21, 2020, which paralleled the period of spikes in the confirmed COVID-19 cases and restricted daily activities. The percentage of uncomplicated and complicated appendicitis (excluding mild appendicitis or normal appendix) in the study group increased between February 26 and March 10, as well as between April 8 and April 21. In the multivariate regression analysis, the odds of uncomplicated and complicated appendicitis increased in three bi-weeks for the study group but not in the control group.

**Data Availability Statement:** All relevant data are within the manuscript.

**Funding:** The author(s) received no specific funding for this work.

**Competing interests:** The authors have declared that no competing interests exist.

## Conclusion

The severity of acute appendicitis might increase during the COVID-19 pandemic, because patients with mild appendicitis (or abdominal pain) may hesitate to seek help.

## Introduction

Coronavirus disease 2019 (COVID-19), a new type of pneumonia, originated in China at the end of 2019, which then rampaged throughout the world, claiming lives and causing economic losses [1]. The World Health Organization declared the COVID-19 epidemic as a pandemic on March 11, 2020 [2,3]. Although there was no effective regimen initially, quarantine, stay-at-home orders and city lockdown were primary countermeasures that various countries adopted to prevent disease dissemination, health system overload, and fatality [4]. Surgery during the pandemic is stressful, as patients and healthcare professionals could be exposed to COVID-19.

Among the situations requiring emergency surgery, surgery for acute appendicitis is one of the most common. Frequent causes of surgery for acute inflammation of the appendix are fecoliths, lymphoid hyperplasia or tumors [5,6]. The number of patients with uncomplicated and complicated appendicitis is unlikely to decrease due to the COVID-19 outbreak. When encountered, appendicitis is frequently managed either by laparoscopic or open appendectomy or occasionally by drainage due to severe adhesions or complications. A decreasing number of patients with appendicitis and an increasing in the incidence of complicated appendicitis have been observed during the COVID-19 outbreak in several countries [4,7–9], while some have not observed any distinct increase in the diagnosis of perforated appendicitis during the pandemic [10]. In addition, there is no information in a relatively contained area (such as Taiwan) regarding the change in the incidence and severity of appendicitis during the COVID-19 pandemic. Therefore, we aimed to investigate the incidence and severity of appendicitis in patients undergoing surgery during the COVID-19 pandemic and compare it with its counterparts from a year ago.

## Materials and methods

### Study population

We retrospectively collected data from patients (study group) who underwent surgery for appendicitis at Taipei City Hospital between January 1, 2020 and June 30, 2020. Patients who underwent surgery for appendicitis at the same hospital between January 1, 2019 and July 1, 2019 (2020 was a leap year) were selected as the control group. Although acute appendicitis can occur throughout the year, some studies have shown that it has a seasonal pattern and is more common during the summer [11]. Hence, the conceptual framework was based on the assumption that the temporal trend of appendicitis during a year was similar. Since the influence of the value of money between two consecutive years could be overlooked (current interest rate < 1% per year), medical expenditure could be compared without considering the net present value. For presenting data, the time frame was divided into 13 bi-week segments starting from January 1, 2020 (study group) or January 1, 2019 (control group). For example, the fifth and tenth bi-week segments represent the intervals from February 26 to March 10 and May 6 to May 19, respectively.

## Independent variables

Patient characteristics included age (years), sex (male/female), comorbidity and the status of the National Health Insurance (NHI). Comorbidity refers to the Deyo version of the Charlson comorbidity index (CCI), which is a composite score derived by assigning different weights to patients whenever they had a prior diagnosis of 17 diseases, including acute coronary syndrome, congestive heart failure, peripheral vascular disease, cerebrovascular accident, dementia, chronic obstructive pulmonary disease, connective tissue disease, non-complicated diabetes mellitus, complicated diabetes mellitus, mild liver disease, moderate or severe liver disease, renal failure, peptic ulcer disease, cancer, metastatic solid tumor, and human immunodeficiency virus infection [12]. The treatment characteristics included the surgery type (conventional open appendectomy, laparoscopic appendectomy, and drainage only), operating time (minutes), anesthesia procedure (epidural/subdural and general), and abdominal wound type (clean, clean-contaminated, and dirty/contaminated). The operating time was calculated from the time of the abdominal wound opening until wound closure. Hospitalization characteristics included the complete hospital stay, hospital stay after surgery, and medical expenditure. As the NHI in Taiwan is a single-payer system provided by the government, the coverage rate usually exceeds 95% in recent decades. Whether the status of NHI influenced the severity of appendicitis is unclear. Therefore, we used this information as an independent variable.

## Dependent variable

The severity of appendicitis could be classified as normal, mild, uncomplicated (moderate), or complicated (severe) appendicitis [13]. Mild appendicitis referred to a hyperemic appendix. Uncomplicated appendicitis referred to marked suppurative appendicitis with or without impending rupture, and appendicitis with local peritonitis. Complicated appendicitis referred to appendicitis with generalized peritonitis, ruptured appendicitis, abscess (tumor formation), sepsis, or septic shock. The dependent variable was a binary indicator of uncomplicated and complicated appendicitis (0 –normal, mild; 1 –uncomplicated, complicated). The Institutional Review Board of the Taipei City Hospital approved this study and waived the requirement for informed consent (no: TCHIRB-11003017-E).

## Statistical analysis

Patient age, operating time, and hospitalization characteristics (including hospital stay and medical expenditure) are reported as means and standard deviations. Patient characteristics (excluding age), treatment (excluding surgery time), and disease variables are reported as numbers and percentages. The chi-square test was used to compare the categorical variables between the study group (during the COVID-19 pandemic) and the control group (1 year prior to the COVID-19 pandemic). The numerical variables (medical expenditure and hospital stay) were compared using Student's t-test. Percentage differences of uncomplicated and complicated appendicitis between the study and control groups were calculated using the following formula:

$$(\% = \frac{number\ of\ uncomplicated\ and\ complicated\ appendicitis\ cases}{number\ of\ normal,\ mild,\ uncomplicated\ and\ complicated\ appendicitis\ cases}; percentage\ difference$$
$$= \%\ of\ the\ study\ group - \%\ of\ the\ control\ group).$$

Univariate and multivariate logistic regression analyses were conducted to assess the possibility of uncomplicated and complicated appendicitis (vs. normal and mild appendicitis) at different periods. Odds ratios (ORs) with corresponding 95% confidence intervals (CIs) were

reported. No adjustment was made for multiple comparisons when the sample size relative to the number of parameters was small. SPSS (version 21.0; SPSS Inc., Chicago, IL, USA) was used for all data management and inferential statistical analyses. All P-values were two-sided, and the significance level was set at P < 0.05.

## Results

In Taiwan, from January 1, 2020 to June 30, 2020, 447 patients had a confirmed diagnosis of COVID-19 and 7 patients died of this disease. Worldwide, during the same period, 20,273,001 patients had a confirmed diagnosis of COVID-19 and 505,295 patients died due to this disease (Fig 1) [14]. The daily confirmed cases of COVID-19 in Taiwan and the world surged after March 20, 2020 and March 25, 2020, respectively. However, this increase lasted only for 1 month in Taiwan. From January 1, 2020 to June 30, 2020 and from January 1, 2019 to July 1, 2019, 307 patients underwent surgery for acute appendicitis (study group: 149 patients; control group: 158 patients). Compared with the control group (70/158, 44.3%), the number of surgeries for appendicitis was relatively low between January 29 and April 21 (49/149, 32.9%) in the study group (Fig 2). The first incision of the abdominal wound was performed between 9:00 PM and 6:00 AM in only 9.4% (29/307) of these patients. Patient characteristics are summarized in Table 1. The age ranged from 6–91 years (mean, 46.2 years). Males comprised 52.1% of all the patients. Most comorbidity indices were 0. Age, sex, and comorbidities were similar between the two periods. Almost all patients (98.0%) received healthcare covered by the NHI,

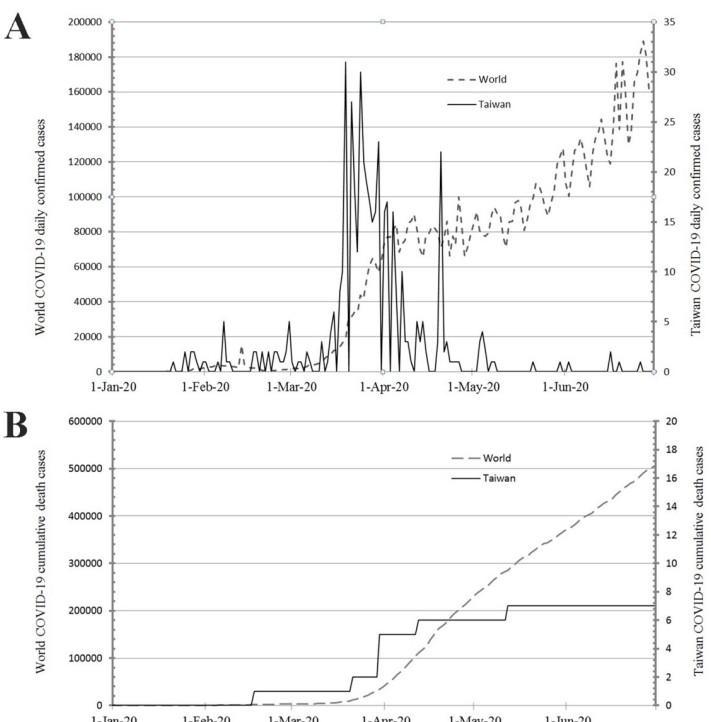

**Fig 1.** A: Daily incidence of coronavirus disease 2019 (COVID-19) globally (dotted line, left scale) and in Taiwan (solid line, right scale). Note the surge in Taiwan began on March 20, 2020, and ended on April 7, 2020 with another short surge on April 20, 2020, while the surge globally began on March 25, 2020, and has still not reached a peak. B: Cumulative deaths due to COVID-19 globally (dotted line, left scale) and in Taiwan (solid line, right scale). Cumulative deaths in Taiwan plateaued after May 12, 2020, while cumulative deaths were still increasing globally [15]. 1-Jan-20 indicates January 1, 2020.

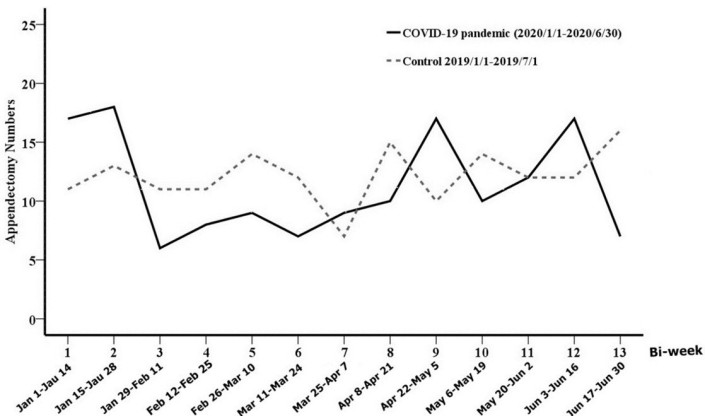

**Fig 2. Temporal trend of the number of surgeries for acute appendicitis in a serial bi-weekly manner (study group: January 1, 2020–June 30, 2020; control group: January 1, 2019–July 1, 2019; leap year: 2020).** The line of the study group depressed from the third bi-week (January 29–February 11) to eighth bi-week (April 8–April 21). Dates are shown for the study group; for the control group, add 1 day to each date after February 28.

a single-payer system in Taiwan. Preoperative abdominal computed tomography (CT) scans were performed in 91.9% (137/149) and 90.5% (143/158) of the patients belonging to the study and control groups, respectively (P = 0.656). Surgical treatment included laparoscopic appendectomy (60.6%, 186/307), conventional appendectomy (38.8%, 119/307) and drainage only (0.7%, 2/307). The percentage of laparoscopic appendectomy was similar in the study (90/149, 60.4%) and control (96/158, 60.8%) groups. Both patients who underwent only drainage belonged to the study group. Surgery was performed under general anesthesia in approximately 80% of the patients. The mean expenditure was 76,517 New Taiwan (NT) dollars (range 25,876–329,085 NT dollars). The median hospital stay in the study group was 3 (range 1–16) days and in the control group 3 (range 1–28) days (P = 0.818).

Mild appendicitis (including normal appendix), uncomplicated appendicitis and complicated appendicitis were found in 36.5% (112/307), 47.6% (146/307), and 16.0% (49/307) of all patients, respectively. Although the percentage of uncomplicated or complicated appendicitis was higher in the study group during the COVID-19 pandemic than that of the control group (68.5% vs. 58.9%), the difference was insignificant (P = 0.213). For comparison, the percentage of uncomplicated or complicated appendicitis over 6 months' period between the study and control groups was 66.9% (103/163). Fecoliths in the appendix were detected in 32.9% (49/149) and 25.3% (40/158) of the patients belonging to the study and control groups, respectively (P = 0.144). The mean operation time was 67 +– 27 min (range: 20–168 min). Dirty or contaminated wounds were observed in a quarter (25.1%, 77/307) of the patients, and there was no intergroup difference. Compared with the control group, the percentage difference of uncomplicated and complicated appendicitis in the study group increased by more than 40% during the fifth bi-week (February 26–March 10) (88.9%, 8/9 vs. the control group 42.9%, 6/14) and eighth bi-week (April 8–April 21) (90.0%, 9/10 vs. the control group 46.7%, 7/15) (Fig 3).

Table 2 summarizes the results of the univariate and multivariate logistic regression analyses of ORs for uncomplicated and complicated appendicitis (vs. normal and mild appendicitis). Univariate analysis of the study group showed that patients with appendicitis undergoing surgery had a higher OR for uncomplicated and complicated appendicitis during February 26–March 10 (OR = 10.00, 95% CI: 1.03–97.50, P = 0.048), April 8–April 21 (OR = 11.25, 95%

**Table 1. Baseline characteristics of patients, treatments, hospitalization for patients undergoing appendectomy.**

| | | | Group | | |
|---|---|---|---|---|---|
| | | All patients | Study group | Control group | |
| | | | 2020/1/1-2020/6/30 | 2019/1/1-2019/7/1 | |
| | | N = 307 | N = 149 | N = 158 | |
| Variables | | n (%), mean +−SD | n (%), mean +−SD | n (%), mean +−SD | P |
| *Patient characteristics* | | | | | |
| **Age** (years) | (range 6–91) | 46.2 +− 19.8 | 45.9 +− 20.0 | 46.4 +− 19.7 | 0.808 |
| **Sex** | | | | | 0.592 |
| | Female | 147 (47.9) | 69 (46.3) | 78 (49.4) | |
| | Male | 160 (52.1) | 80 (53.7) | 80 (50.6) | |
| **Comorbidity** | | | | | 0.617 |
| | 0 | 269 (87.6) | 132 (88.6) | 137 (86.7) | |
| | > = 1 | 38 (12.4) | 17 (11.4) | 21 (13.3) | |
| **NHI coverage** | | | | | 0.369 |
| | No | 6 (2.0) | 4 (2.7) | 2 (1.3) | |
| | Yes | 301 (98.0) | 145 (97.3) | 156 (98.7) | |
| *Treatment characteristics* | | | | | |
| **Abdominal CT** | | | | | 0.656 |
| | No | 27 (8.8) | 12 (8.1) | 15 (9.5) | |
| | Yes | 280 (91.2) | 137 (91.9) | 143 (90.5) | |
| **Fecolith** | | | | | 0.144 |
| | No | 218 (71.0) | 100 (67.1) | 118 (74.7) | |
| | Yes | 89 (29.0) | 49 (32.9) | 40 (25.3) | |
| **Severity of appendicitis** | | | | | 0.213 |
| | Mild/normal | 101/11 (36.5) | 42/5 (31.5) | 59/6 (41.1) | |
| | Uncomplicated | 146 (47.6) | 77 (51.7) | 69 (43.7) | |
| | Complicated | 49 (16.0) | 25 (16.8) | 24 (15.2) | |
| **Surgery type** | | | | | 0.343 |
| | Laparoscopic appendectomy | 186 (60.6) | 90 (60.4) | 96 (60.8) | |
| | Open appendectomy | 119 (38.8) | 57 (38.3) | 62 (39.2) | |
| | Drainage only | 2 (0.7) | 2 (1.3) | 0 (0.0) | |
| **Anesthesia** | | | | | 0.196 |
| | General | 244 (79.5) | 123 (82.6) | 121 (76.6) | |
| | Epidural/subdural | 63 (20.5) | 26 (17.4) | 37 (23.4) | |
| **Wound type** | | | | | 0.333 |
| | Dirty or contaminated | 77 (25.1) | 40 (26.8) | 37 (23.4) | |
| | Clean contaminated | 114 (37.1) | 59 (39.6) | 55 (34.8) | |
| | Clean | 116 (37.8) | 50 (33.6) | 66 (41.8) | |
| **Operation time** (minutes) | (range 20–168) | 67 +− 27 | 68 +− 29 | 66 +− 25 | 0.590 |
| *Hospitalization characteristics* | | | | | |
| **Hospital stay** (days) | (range 1–28) | 5.0 +− 3.2 | 5.0 +− 2.9 | 4.9 +− 3.5 | 0.992 |
| **Hospital stay** (days, after surgery only) | (range 1–28) | 4.4 +− 3.1 | 4.4 +− 2.9 | 4.5 +− 3.4 | 0.818 |
| | median, IQR | 3, 3–5 | 3, 2–5 | 3, 3–5 | |
| **Medical expenditure** (NT dollars) | (range 25876–329085) | 76517 +− 38842 | 77608 +− 36767 | 75487 +− 40792 | 0.633 |
| | median, IQR | 65698, 51212–92114 | 67060, 52175–91425 | 63066, 50980–92397 | |

N, n: Number; SD: Standard deviation; comorbidity index codified according to Charlson Comorbidity Index (Deyo version); NHI: National Health Insurance (Taiwan); CT: Computed tomography; NT dollars: New Taiwan dollars; 1 US dollar = 30 NT dollars); IQR: Interquartile range.

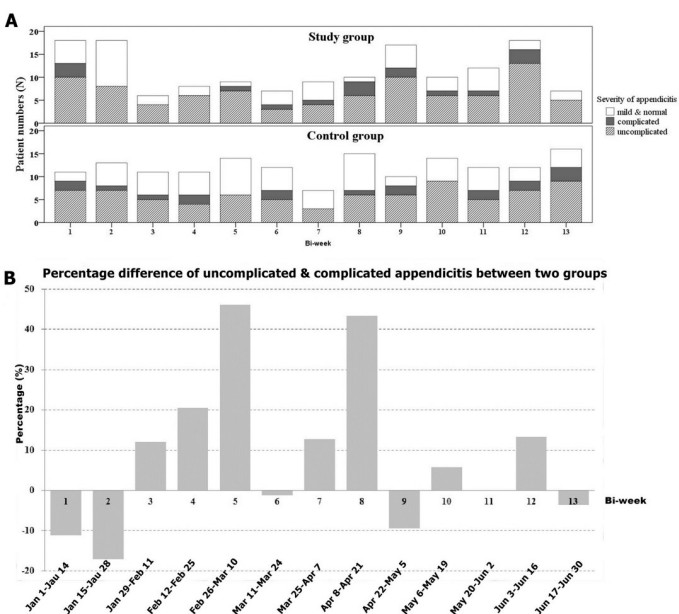

**Fig 3.** A: Temporal trend of the severity of appendicitis in patients undergoing surgery during the coronavirus disease 2019 (COVID-19) outbreak (study group: Upper row; control group: Lower row). The number of mild appendicitis cases was low during the third to fifth bi-week (January 29–March10) and the eighth bi-week (April 8–April 21) of the study group. B: Percentage difference of uncomplicated and complicated appendicitis between the study and control groups. The percentage of uncomplicated and complicated appendicitis in patients undergoing surgery increased during the COVID-19 outbreak in the fifth bi-week (February 26–March 10) and eighth bi-week (April 8–April 21) in the study group. Dates are shown for the study group; for the control group, add 1 day to each date after February 28.

CI: 1.17–108, P = 0.036), and June 3–June 16 (OR = 10.00, 95% CI: 1.76–56.93, P = 0.009). Multivariate analysis of the study group showed that patients with appendicitis undergoing surgery had a higher OR for uncomplicated and complicated appendicitis during January 15–January 28 (OR = 5.81, 95% CI: 1.18–28.59, P = 0.030), February 26–March 10 (OR = 12.63, 95% CI: 1.07–149, P = 0.044), and June 3–June 16 (OR = 13.46, 95% CI: 1.98–91.56, P = 0.008). However, such a significant increase in OR (bi-weekly) for the severity of appendicitis was not observed in the univariate or multivariate analyses of the control group. During the COVID-19 pandemic, laparoscopic appendectomy was associated with shorter hospital stay after surgery (P = 0.011) and lower medical expenditure (P = 0.035) when compared with open appendectomy; similar findings (excluding expenditure) were observed during the control period (Table 3).

## Discussion

Many areas in Asia were less impacted by COVID-19 than other areas. Possibly due to the mandatory mask-wearing policy and social distancing norms early on in the pandemic, the impact of COVID-19 in Taiwan was relatively temporary and limited compared with other parts of the world. However, we noted that the number of surgeries for appendicitis might have decreased during the COVID-19 pandemic compared with a similar period a year before the pandemic. This probably occurred when the number of confirmed cases of COVID-19 surged in Taiwan. In addition, the severity of appendicitis also increased, which appeared to inversely parallel the number of surgeries for appendicitis during the pandemic. Patients with self-limiting lower abdominal pain might be reluctant to seek help from healthcare providers

**Table 2. Univariate and multivariate logistic regression analyses of odds ratios for uncomplicated and complicated appendicitis (vs. normal and mild appendicitis).**

| | Study group (2020) | | | | Control group (2019) | | | |
|---|---|---|---|---|---|---|---|---|
| | Univariate | | Multivariate | | Univariate | | Multivariate | |
| | OR (95% CI) | P value | OR (95% CI) | P value | OR (95% CI) | P value | OR (95% CI) | P value |
| Bi-week interval (Jan 1–Jan 14)* | | 0.389 | | 0.379 | | 0.616 | | 0.660 |
| Jan 15–Jan 28, bi-week 2 | 3.25 (0.81–13.03) | 0.096 | 5.81 (1.18–28.59) | 0.030 | 2.81 (0.42–18.74) | 0.285 | 6.38 (0.75–54.14) | 0.090 |
| Jan 29–Feb 11, bi-week 3 | 2.50 (0.36–17.32) | 0.353 | 3.51 (0.41–29.79) | 0.250 | 0.75 (0.15–3.83) | 0.729 | 1.22 (0.18–8.32) | 0.838 |
| Feb 12–Feb 25, bi-week 4 | 3.75 (0.59–23.87) | 0.162 | 7.18 (0.91–56.43) | 0.061 | 0.75 (0.15–3.83) | 0.729 | 1.51 (0.24–9.65) | 0.665 |
| Feb 26–Mar 10, bi-week 5[#] | 10.00 (1.03–97.50) | 0.048 | 12.63 (1.07–149) | 0.044 | 0.47 (0.10–2.18) | 0.335 | 0.60 (0.10–3.61) | 0.579 |
| Mar 11–Mar 24, bi-week 6[#] | 1.67 (0.29–9.71) | 0.570 | 1.13 (0.16–8.02) | 0.906 | 0.87 (0.18–4.34) | 0.870 | 1.71 (0.27–10.79) | 0.569 |
| Mar 25–Apr 7, bi-week 7[#] | 1.56 (0.31–7.82) | 0.587 | 2.66 (0.43–16.29) | 0.291 | 0.47 (0.07–3.04) | 0.427 | 0.79 (0.09–7.26) | 0.837 |
| Apr 8–Apr 21, bi-week 8[#] | 11.25 (1.17–108) | 0.036 | 9.46 (0.80–111.99) | 0.075 | 0.55 (0.12–2.47) | 0.433 | 1.33 (0.23–7.62) | 0.747 |
| Apr 22–May 5, bi-week 9[#] | 3.00 (0.74–12.13) | 0.123 | 4.51 (0.86–23.79) | 0.076 | 2.50 (0.37–16.89) | 0.347 | 2.29 (0.27–19.6) | 0.449 |
| May 6–May 19, bi-week 10[#] | 2.92 (0.57–15.05) | 0.201 | 5.80 (0.87–38.67) | 0.069 | 1.13 (0.24–5.37) | 0.883 | 2.62 (0.44–15.75) | 0.293 |
| May 20–June 2, bi-week 11[#] | 1.75 (0.40–7.66) | 0.458 | 3.84 (0.72–20.43) | 0.115 | 0.87 (0.18–4.34) | 0.870 | 1.71 (0.27–10.95) | 0.571 |
| June 3–June 16, bi-week 12[#] | 10.00 (1.76–56.93) | 0.009 | 13.46 (1.98–91.56) | 0.008 | 1.87 (0.34–10.46) | 0.474 | 5.02 (0.68–36.89) | 0.113 |
| June 17–June 30, bi-week 13[#] | 3.13 (0.47–20.58) | 0.236 | 4.25 (0.53–34.10) | 0.173 | 1.87 (0.38–9.20) | 0.438 | 2.92 (0.47–18.11) | 0.251 |

OR: Odds ratio, CI: Confidence interval;

*: Interval in the parentheses indicate reference group; estimates were derived after controlling age, sex, comorbidity and surgery type;

#: Add 1 day to each date after February 28 in the control group.

to avoid contracting COVID-19. Therefore, during the COVID-19 pandemic, the number of surgeries for mild appendicitis seemed to decrease, and the severity of appendicitis increase.

In more severely COVID-19 infected areas, researchers have observed delays in consultation, higher rates of severe peritonitis and higher rates of complicated appendicitis in the pandemic cohort [15,16]. Researchers in Colombia have demonstrated that the severity of appendicitis was higher during the pandemic (92%) than in the same period a year before the pandemic (57.1%, P = 0.003) [4]. They suggested that patients with mild abdominal pain during the pandemic might postpone visiting the emergency room until their symptoms become irreversible. A study in China reported a higher proportion of ruptured appendicitis during the pandemic when compared with that in 2019 [17]. For patients who were quarantined for COVID-19, a study reported 44.9% (48/107) of patients who presented with acute appendicitis, and 33.3% (16/48) of these patients with acute appendicitis had a ruptured appendix [18]. However, there have been different reports. In a 7-week interval study conducted in Israel, although the incidence of acute appendicitis showed a significant decrease during the COVID-19 pandemic, the percentage of complicated appendicitis did not show a significant difference

**Table 3. Medical expenditure and hospital stay of patients undergoing appendectomy according to surgery type.**

| | Study group | | | Control group | | |
|---|---|---|---|---|---|---|
| | 2020/1/1-2020/6/30 | | | 2019/1/1-2019/7/1 | | |
| Variables | Open appendectomy | Laparoscopic appendectomy | | Open appendectomy | Laparoscopic appendectomy | |
| | N = 57 | N = 90 | P | N = 62 | N = 96 | P |
| Medical expenditure (NT dollars) | 81236 +− 47942 | 75361 +− 27762 | 0.035 | 80090 +− 53694 | 72514 +− 29598 | 0.256 |
| Hospital stay (days, after surgery only) | 5.1 +− 2.8 | 3.9 +− 2.8 | 0.011 | 5.5 +− 4.1 | 3.8 +− 2.7 | 0.002 |

N: Number; NT dollars: New Taiwan dollars.

between the study and control groups [7]. Additionally, another survey conducted in Turkey did not observe any increase in perforated appendicitis during the pandemic [10]. The normal appendix rate was estimated to be 3.6% (11/307) in our study, which was comparable to other reports. The average rate has been reported to be 20.6% in large, international multicenter studies and as low as 3.3% and as high as 36.8% in individual centers [19]. The lower rate of our result could be partially attributed to the high coverage of the NHI and no restrictions with respect to abdominal or pelvic CT before surgery. A high abdominal CT examination rate (91.2%) before surgery might help surgeons assess the severity and subsequently decrease the rate of normal appendices.

Abdominal CT has often been used as a useful diagnostic tool for acute appendicitis; however, it should not be advocated during the COVID-19 pandemic. It could present a logistical challenge and increase the risk of cross-exposure of the staff and patients when we should be aiming to minimize this [20]. A reduction in the number of abdominal CT scans used to detect early or mild acute appendicitis could be expected during the COVID-19 pandemic. Thus, those diagnosed with appendicitis by CT presented at later stages with greater disease severity [4]. When conservative treatment is considered, it is necessary to confirm the diagnosis by imaging techniques, such as abdominal CT [21]. Apart from identifying fluid accumulation or an abscess around the appendix, CT may also visualize fecoliths in the appendix. Although the presence of a fecolith in the appendix did not necessarily indicate appendicitis, approximately one-third of the patients with appendicitis presented with a fecolith [22], which was similar to our results. The presence of fecoliths increases the likelihood of appendiceal perforation.

The ability of surgical services to maintain a role depends on the peak of the pandemic, the spread of the disease, the duration of societal regulations, and the duration and temporal epidemic periods by which the disease burden will approach the threshold of maximum capacity of the critical care resources [23]. During the COVID-19 pandemic, several national surgical societies and organizations have issued guidance for surgery including the role of conservative management and concerns over laparoscopic surgery [24–27]. Studies have indicated that the management of acute appendicitis has shifted to more conservative interventions during the pandemic. In the UK, the COVID-19 outbreak has changed the treatment of appendicitis with a conservative method which has proven to be effective [28,29]. Regarding uncomplicated and complicated acute appendicitis, a meta-analysis comparing appendectomy (N = 1,288) and conservative treatment (N = 1,463) found that conservative treatment with antibiotics was significantly associated with fewer complications and shorter hospitalization; however, it had a higher relapse rate [30]. Nonetheless, the surgical community frequently expresses concerns about the seemingly high rates of failure, and the risk of prolonged hospitalization [31]. Furthermore, a report showed that even in a patient with confirmed COVID-19 undergoing appendectomy, severe acute respiratory syndrome coronavirus 2 was not detected in the peritoneal fluid or peritoneal washings [32].

The surgical technique (laparoscopic or open appendectomy) for treating patients with appendicitis during the COVID-19 pandemic is also debatable. Although laparoscopic appendectomy comprised approximately 60% of all appendectomies in the current study, it was not advocated by some investigators [24]. The potential of aerosol-borne viruses released by laparoscopy in a COVID-19 positive patient might expose the personnel in the operating room to the risk of contamination [2]. A study conducted in the UK showed that appendectomy during the pandemic (56% open appendectomy) was significantly associated with practice before the pandemic (0.4% open appendectomy) due to guidance recommendations [28]. The route of contamination in operating rooms through aerosol is particularly important in relation to the evacuation of the pneumoperitoneum at the end of a laparoscopic surgery [33]. In the USA and elsewhere in Europe, specialty bodies did not condemn laparoscopic surgery [34]. A

survey of 744 surgeons (from 66 countries) revealed that only one-third changed their approach from laparoscopic to open surgery owing to the popular (but evidence-lacking) advice from expert groups [35]. Obviously, there are discordant attitudes globally toward laparoscopic techniques in appendectomy during the pandemic.

The present study had certain limitations. First, the COVID-19 pandemic in Taiwan was relatively limited during the first half of the year compared with other countries. The spikes in confirmed COVID-19 cases lasted less than 2 months and subsequent lockdown of social activities including access to healthcare facilities persisted for less than 4 months. The condition of patients with mild abdominal pain who reluctantly sought medical help due to the COVID-19 pandemic might not resemble the condition that most countries confronted. Exhaustion of health capacity caused by COVID-19 was not seen or was mild, if any. Second, as noted by some investigators, it took about 4 weeks after the onset of COVID-19 for the change in the incidence of acute appendicitis to become apparent [7]; therefore, the magnitude of the change in the incidence in the current study might not match that of other countries. Through different study designs (bi-week) in the current study, changes in the incidence of appendicitis could still be observed. However, this scale inflated the number of regression models, leading to potentially erroneous inferences. Therefore, the results should be carefully interpreted. Third, studies investigating the incidence of surgery usually suffered from representativeness, as patients could seek any hospitals they preferred instead of considering the location. However, with the pandemic restriction, patients with acute appendicitis usually sought hospitals located in their neighborhood as people could not travel freely. Finally, the selection of the comparison period was arbitrary. We were unsure which period was the best for the control group; however, we assumed that the ideal control group might be a calendar year ahead of the pandemic, wherein the incidence, severity, and type of surgery for acute appendicitis could be assumed to be equivalent.

## Conclusions

In conclusion, COVID-19 has disrupted healthcare systems, practices, and accessibility globally. The number of patients with acute appendicitis seeking surgical treatment might decrease as confirmed cases of COVID-19 surged, even in areas where the outbreak was relatively mild. We noted that the severity of appendicitis was higher during periods with fewer appendectomies, which suggests that patients with mild appendicitis (or abdominal pain) might reluctantly seek help until symptoms become aggravated. In Taipei, strategies of mask-wearing and social distancing contained the outbreak of COVID-19 during the first half of the year, therefore, the influence on the incidence and severity of appendicitis during this period was not comprehensive.

## Author Contributions

**Conceptualization:** Yao-Jen Chang.

**Data curation:** Li-Ju Chen, Yun-Jau Chang.

**Formal analysis:** Yun-Jau Chang.

**Investigation:** Yao-Jen Chang, Li-Ju Chen.

**Methodology:** Li-Ju Chen, Yun-Jau Chang.

**Resources:** Li-Ju Chen.

**Software:** Yun-Jau Chang.

**Supervision:** Yao-Jen Chang.

**Validation:** Yao-Jen Chang.

**Visualization:** Li-Ju Chen, Yun-Jau Chang.

**Writing – original draft:** Yun-Jau Chang.

**Writing – review & editing:** Li-Ju Chen.

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
