## [Decision Letter · Decision Letter 0]

18 May 2021

PONE-D-21-07736

The severity of appendicitis increases during the COVID-19 pandemic?

PLOS ONE

Dear Dr. Chang,

Thank you for submitting your manuscript to PLOS ONE. After careful consideration, we feel that it has merit but does not fully meet PLOS ONE’s publication criteria as it currently stands. Therefore, we invite you to submit a revised version of the manuscript that addresses the points raised during the review process.

We look forward to receiving your revised manuscript.

Kind regards,

Chun Chieh Yeh, M.D., Ph.D.

Academic Editor

PLOS ONE

Additional Editor Comments:

Your work is interesting. Please make timely revision based on our referees' comments.

Journal Requirements:

Reviewers' comments:

Reviewer's Responses to Questions

**Comments to the Author**

1. Is the manuscript technically sound, and do the data support the conclusions?

Reviewer #1: Partly

Reviewer #2: Partly

2. Has the statistical analysis been performed appropriately and rigorously? 

Reviewer #1: No

Reviewer #2: No

3. Have the authors made all data underlying the findings in their manuscript fully available?

Reviewer #1: Yes

Reviewer #2: No

4. Is the manuscript presented in an intelligible fashion and written in standard English?

Reviewer #1: No

Reviewer #2: No

5. Review Comments to the Author

Reviewer #1: Comments to the Author PONE-D-21-07736

This is an interesting article discussing the the severity of appendicitis during the COVID-19 pandemic. The authors assume that the timing of seeking medical help in patients with appendicitis may potentially procrastinate during the pandemic. They conclude that the severity of acute appendicitis might increase during pandemic, probably due to decrease attitude to seek help in patient with mild appendicitis (or abdominal pain).

I have several comments,

1. Suggest that the title should have a clear, precise scientific meaning and should not be phrased as a question. Where possible, the title should be read as one concise sentence.

2. Is the severity of appendicitis based on clinical finding or on pathologic report?

3. Previous studies had already showed an increasing incidence of complicated appendicitis during pandemic, which might be due to that patients are not seeking appropriate, timely surgical care (As your references 9, 10, 11…etc.). Moreover, there were some other similar studies such as (1. Lee-Archer P, et al. J Paediatr Child Health. 2020 Aug; 56(8):1313-1314. 2. Cano-Valderrama O, et al. Int J Surg. 2020 Aug; 80:157-161.) . Therefore, the finding in this article may not be original. Additionally, the conclusion seems not be robustly deduced from the result. For a confined evidence rather than a plausible assumption, suggest to search or establish a solid indicator(s) for the association of quarantine/mask policy and the increased severity of appendicitis.

4. Why choose a “bi-week” scale rather than a “month” scale?

5. In the method section, the study period for study group is January 1, 2020 and June 30, 2020, while for control group is January 1, 2019 and July 1, 2019 (leap year for 2020). However, the study group is 2020/1/1-2020/6/30 and the control group is 2019/1/1-2019/6/30 in table 1. May amend and check the exact date among the figures and tables.

6. Need to add “minutes” to the operation time in table 1.

7. Suggest performing a comparison of study group to control group in table 2. An alternative is to perform an interaction test to ascertain the differences between two groups

8. Please add related description of figure 1 and 2 in the result section. Additionally, why there were data of 2019/7/2-2019/12/30 in figure 2, 3?

9. How to measure the “Temporal change” in figure 3B? There was no related description in the method section.

10. The section of discussion is a bit long and need concise.

11. The style of references is inconsistent and confusing.

12. Would encourage the authors to obtain assistance in English writing.

Reviewer #2: This is an interesting study that compares the grade and overall number of presentations with appendicitis at a Taiwanese hospital during two time periods – the first 6 months of 2019 and the first 6 months of 2020 (i.e. during the COVID-19 pandemic). The authors hypothesize that concern about the pandemic impacted treatment-seeking behavior in the populace, in turn leading to delay in seeking medical treatment and thus increased severity at eventual presentation.

I have several major concerns about the methodology employed in this study, as outlined below.

1. The statistical analysis section is lacking details required to fully understand how the authors obtained their results. For example, there are no details on any univariable analyses as per the results seen in Table 2 or the subgroup analyses to obtain the results in Table 3. I suggest the authors revise this section to include these details, in addition to briefly outlining how the figures were produced.

2. Patient presentation data are divided into thirteen “bi-week” periods to capture change in presentations over time using logistic regression. However, this approach inflates the number of regression models required and dilutes the sample size being examined in each of these statistical models. This can lead to potentially erroneous inferences (i.e. the multiple comparisons problem) and imprecise odds ratio estimates, as evidenced by the extremely large 95% confidence intervals in Table 2. This is a serious limitation of the current approach which is not currently discussed in the manuscript.

I recommend the authors provide major revisions to the results and discussion to address these issues. Suggested approaches to consider would include an overall analysis comparing severe appendicitis presentations in the study and control groups, broadening of the time periods considered (e.g. before, during and after the main outbreak in Taiwan), or a time-series analysis to estimate a moving average of severe appendicitis cases over time. If the authors also decide to retain the current results, they should adjust for multiple comparisons and revise the language to describe the results so as not to overstate their findings.

3. Further to point 2, Table 2 implies that the reference interval is the first two weeks of each year. The odds ratios produced for the study group describe the relative change in odds of severe appendicitis as compared to the first week of that year (i.e. bi-week 8 of 2020 compared to bi-week 1 of 2020), which is then indirectly used to compare study populations. A more intuitive approach would be directly comparing corresponding intervals between years (i.e. bi-week 8 of 2020 compared to bi-week 8 of 2019), especially as the authors describe seasonal patterns in appendicitis presentations. If the authors maintain the "bi-week" analytical approach, I advise they include further explanation of why this method was chosen to make it clear why the direct comparison was not used.

4. The language to describe severity of appendicitis is not used consistently throughout. The severity categories are “normal, mild, uncomplicated and complicated” in some places (e.g. the Dependent variables section), and “normal, mild, moderate or severe” in others (e.g. Abstract and Results section). Clarification of these categories will improve readability. I strongly suggest clarifying these terms and explicitly defining the outcome of interest for the logistic regression analysis in the Dependent variable section.

5. I suggest avoiding using the word “significant” when reporting differences, as there is evidence that the p-value significance criteria of <0.05 is too high and contributing to the reproducibility crisis in research (see Benjamin, D.J., Berger, J.O., Johannesson, M. et al. Redefine statistical significance. Nat Hum Behav 2, 6–10 (2018).

6. “Multivariate logistic regression analysis was conducted to assess the possibility of getting ruptured (or impending) ruptured appendicitis at different time period”. This does not clearly define the outcome of interest using the established appendicitis severity ratings, and does not describe the groups being compared, and needs to be revised for clarity

7. “Severity of acute appendicitis might increase during COVID-19 pandemic, probably due to decrease attitude to seek help in patient with mild appendicitis (or abdominal pain)”. The wording is very strong and risks overstating findings. The use of “probably” should be substituted with more cautious wording, such as “potentially” or “may be”.

8. I would recommend the authors include a limitations section in their Discussion section to highlight any caveats on their conclusions.

I also have some suggestions regarding the tables and figures in the manuscript:

9. Table 1 suggestions:

- Include subheadings to indicate which characteristics relate to patients, treatments and hospitalization.

- Fecolith should have a “No” category.

- Consider reporting median and interquartile range for some continuous outcomes such as cost and length of hospital stay. This is to be consistent with the Results section (“Median hospital stay in the study group was 3 (1–16) days, vs. 3 (1–28) days in the control group (p = 0.818)”) and because these variables are often non-normally distributed, so the median and interquartile range better capture the spread of the variable.

-In the Statistical analysis section, include detail about method used to get the p-value comparing continuous outcomes in Table 1

10. Table 2 caption should include details about univariable analysis and what the comparator groups are.

11. Table 3 needs an appropriate caption to indicate stratification by surgery type.

12. For Figure 1, the relevance of world COVID-19 case and fatality figures is not clear. Do the authors anticipate rising cases around the world would have some impact on hospital attendance in Taiwan in the absence of local cases? E.g. Was there ongoing concern about possible imported cases?

13. For Figure 2, I suggest that the x-axis include dates ranges instead of bi-week number for easier comparison with the case numbers provided by date in Figure 1. The reason for including the “reference” line (July 2019-December 2020) is not clear, as this time period was not discussed in the manuscript at all – could the authors please justify this?

14. Figure 3 suggestions:

- The use of percentage differences should be replaced with actual raw value differences in B, as percentages can be misleadingly large when there are few appendicitis cases.

- As with Figure 2, suggest labelling each bi-week with the corresponding date range for easier comparison with the case numbers shown in Figure 1, as well as discussion of why the July 2019-December 2020 timeframe was included.

6. PLOS authors have the option to publish the peer review history of their article (what does this mean?). If published, this will include your full peer review and any attached files.

Reviewer #1: No

Reviewer #2: No

---

## [Author Response · Author response to Decision Letter 0]

2 Jul 2021

Author’s point-by-point response to review comments (PONE-D-21-07736) : 

We have carefully revised the manuscript in view of the editorial and reviewers’ comments as outlined in detail below.

Reviewer’s comments:

Reviewer #1: I have several comments,

1. Suggest that the title should have a clear, precise scientific meaning and should not be phrased as a question. Where possible, the title should be read as one concise sentence. Response: Yes, we thank the reviewer’s comment. The title showed the question we wanted to know whether the severity of appendicitis increased during the COVID-19 pandemic, especially in Taiwan, where situation of COVID-19 seemed relatively milder than other countries for the first half year. After all, the results indicated the severity increased only in the short term, instead of all the half year. We had no robust result that the severity of appendicitis increased all the half year, that is why the title is rephrased as a question “Did the severity of appendicitis increase during the COVID-19 pandemic?”. Because COVID-19 is an emerging disease, some research such as References 11, 32, 35 might use questionable titles to express this uncertainty.

2. Is the severity of appendicitis based on clinical finding or on pathologic report? Response: Yes, thanks to reviewer’s comment. The severity of appendicitis was based mainly on pathologic report except for those who had no specimen examined. Clinical finding (such as CT report) was also important to classify the severity. 

3. Previous studies had already showed an increasing incidence of complicated appendicitis during pandemic, which might be due to that patients are not seeking appropriate, timely surgical care (As your references 9, 10, 11…etc.). Moreover, there were some other similar studies such as (1. Lee-Archer P, et al. J Paediatr Child Health. 2020 Aug; 56(8):1313-1314. 2. Cano-Valderrama O, et al. Int J Surg. 2020 Aug; 80:157-161.) . Therefore, the finding in this article may not be original. Additionally, the conclusion seems not be robustly deduced from the result. For a confined evidence rather than a plausible assumption, suggest to search or establish a solid indicator(s) for the association of quarantine/mask policy and the increased severity of appendicitis. 

Reply: Thanks to reviewer’s comment. Indeed, there were studies using various methods to examine the increasing incidence of complicated (perforated) appendicitis during the pandemic, but not all studies drew the same conclusion. At least one research did not observe any clear increase in the diagnosis of perforated appendicitis during the pandemic period [1, ref 11 of the manuscript]. In our present study, we also did not observe the severity of appendicitis get increased for all the half year. We found the severity of appendicitis got increased only in certain period (within period of spikes of COVID-19 cases). In addition, for neither single indicator such as complicated (or perforated) nor uncomplicated (moderate or non-perforated) appendicitis, did we find significant increase of numbers during the COVID-19 pandemic. Instead, decreased percentage of mild appendicitis could be observed (as shown in Figure 3A). The reasonable explanation was that patients with self-limiting lower abdominal pain might be reluctant to seek help from healthcare providers to avoid contracting COVID-19. Therefore, we presented our results in our present study. 

Reference 

1. Turanli S, Kiziltan G: Did the COVID-19 Pandemic Cause a Delay in the Diagnosis of Acute Appendicitis? World J Surg 2020. 45 (1), 18–22.

4. Why choose a “bi-week” scale rather than a “month” scale? Response: We would like to thank the reviewers for thoughtful comment. A “bi-week” scale represents 14 days, but there are four types of possible lengths for a “month” (28, 29, 30, 31 days), which just all occurred during our study period. We thought a “bi-week” had stable length rather than a “month”, especially when involving a leap year. In addition, the spike and plateau of the confirmed numbers in our study exceeded no more than one month during the 6 months (refers to Figure 1). Use the scale of a “month” might not explicitly observe the change of the severity of appendicitis. So, for a country like Taiwan where the pandemic was not consistently overwhelmed for more than three months, we chose a “bi-week” scale to better inspect this change.

5. In the method section, the study period for study group is January 1, 2020 and June 30, 2020, while for control group is January 1, 2019 and July 1, 2019 (leap year for 2020). However, the study group is 2020/1/1-2020/6/30 and the control group is 2019/1/1-2019/6/30 in table 1. May amend and check the exact date among the figures and tables. Response: Yes, we thank the reviewer’s comment. We have made a correction of Table 1 in the revised manuscript. 

6. Need to add “minutes” to the operation time in table 1. Response: Yes, we thank the reviewer’s comment. We have made a change accordingly in the revised manuscript.

7. Suggest performing a comparison of study group to control group in table 2. An alternative is to perform an interaction test to ascertain the differences between two groups. Response: Yes, we thank the reviewer’s comment. Although a comparison of study group to control group in table 2 was suggested, the P value of the variable “Severity of appendicitis” could be seen as 0.213 in Table 1 (using Chi-Square test). It indicated that the severity of appendicitis did not significantly increase during all the six months at Taipei City Hospital. Further investigation using this categorization in these first six months would reveal similar data. Therefore, it was needed to utilize smaller scale (such as a “bi-week” or a “month” scale) to inspect this change. 

8. Please add related description of figure 1 and 2 in the result section. Additionally, why there were data of 2019/7/2-2019/12/30 in figure 2, 3? Response: 

A. For Figure 1, we had related description in the result section already. “In Taiwan, Ffrom January 1, 2020 to June 30, 2020, there were 447 patients who had a confirmed diagnosis of COVID-19 and 7 patients died due to of this disease in Taiwan. Worldwide, during At the same period, there were 20,273,001 patients who had a confirmed diagnosis of COVID-19 and 505,295 patients died due to this diagnosis disease all over the world (Figure 1) [15]. The daily confirmed cases of COVID-19 in Taiwan and the world surged after March 20, 2020 and March 25, 2020, respectively. However, such this increase lasted only for one 1 month in Taiwan.” (1st, 2nd, 3rd and 4th sentences, 1st paragraph, Results section). B. For Figure 2, we had related description in the result section already. “Compared to with the control group, the number of surgery surgeries for appendicitis was relatively low between January 29 and March 24 April 21 in the study group (Figure 2).” (6th sentence, 1st paragraph, Results section). C. We thank the reviewer’s comment. We delete data of 2019/7/2-2019/12/30 in figures 2 and 3 in the revised manuscript according to reviewer’s suggestion.

9. How to measure the “Temporal change” in figure 3B? There was no related description in the method section. Response: Yes, we thank the reviewer’s comment. We have added a descrpition in the revised manuscript. “Percentage differences in uncomplicated and complicated appendicitis between the study and control groups were calculated using the following formula: ( ; ).” (5th sentence, 4th paragraph, Materials and methods section)

10. The section of discussion is a bit long and need concise. Response: Yes, we thank the reviewer’s comment. We have made section of discussion a shorter change in the revised manuscript (especially in 2nd paragraph of Discussion)

11. The style of references is inconsistent and confusing. Response: Yes, we thank the reviewer’s comment. We have made several changes (adding pages number) of references ([1], [9], [10], [11], [16], [19], [21], [24], [27], [28], [29], [30], [36]) in the revised manuscript. 

12. Would encourage the authors to obtain assistance in English writing. Response: Yes, we thank the reviewer’s comment. We have obtained assistance in English writing in the revised manuscript.

Reviewer #2: I have several major concerns about the methodology employed in this study, as outlined below, 

1. The statistical analysis section is lacking details required to fully understand how the authors obtained their results. For example, there are no details on any univariable analyses as per the results seen in Table 2 or the subgroup analyses to obtain the results in Table 3. I suggest the authors revise this section to include these details, in addition to briefly outlining how the figures were produced. Response: Yes, we thank the reviewer’s comment. We have made two changes in the revised manuscript. “The numerical variables (medical expenditure and hospital stay) were compared using Student’s t-test. Percentage differences in uncomplicated and complicated appendicitis between the study and control groups were calculated using the following formula: ( ; ). Univariate and Mmultivariate logistic regression analysies was were conducted to assess the possibility of getting ruptured (or impending) ruptured uncomplicated and complicated appendicitis (vs. normal & mild appendicitis) at different time periods” (4th, 5th and 6th sentences, 4th paragraph, Materials and methods section).

2. Patient presentation data are divided into thirteen “bi-week” periods to capture change in presentations over time using logistic regression. However, this approach inflates the number of regression models required and dilutes the sample size being examined in each of these statistical models. This can lead to potentially erroneous inferences (i.e. the multiple comparisons problem) and imprecise odds ratio estimates, as evidenced by the extremely large 95% confidence intervals in Table 2. This is a serious limitation of the current approach which is not currently discussed in the manuscript.

I recommend the authors provide major revisions to the results and discussion to address these issues. Suggested approaches to consider would include an overall analysis comparing severe appendicitis presentations in the study and control groups, broadening of the time periods considered (e.g. before, during and after the main outbreak in Taiwan), or a time-series analysis to estimate a moving average of severe appendicitis cases over time. If the authors also decide to retain the current results, they should adjust for multiple comparisons and revise the language to describe the results so as not to overstate their findings. Response: Yes, we thank the reviewer’s comment. We have added a limitation section in the revised manuscript. “Second, as noted by some investigators, it took about 4 weeks after the onset of COVID-19 for the change in the incidence of acute appendicitis to become apparent [7]; therefore, the magnitude of the change in the incidence in the current study might not match that of other countries. Through different study designs (bi-week) in the current study, changes in the incidence of appendicitis could still be observed. However, this scale inflated the number of regression models, leading to potentially erroneous inferences. Therefore, the results should be careful interpreted.” (6th, 7th, 8th and 9th sentences, limitation paragraph, Discussion section)

3. Further to point 2, Table 2 implies that the reference interval is the first two weeks of each year. The odds ratios produced for the study group describe the relative change in odds of severe appendicitis as compared to the first week of that year (i.e. bi-week 8 of 2020 compared to bi-week 1 of 2020), which is then indirectly used to compare study populations. A more intuitive approach would be directly comparing corresponding intervals between years (i.e. bi-week 8 of 2020 compared to bi-week 8 of 2019), especially as the authors describe seasonal patterns in appendicitis presentations. If the authors maintain the "bi-week" analytical approach, I advise they include further explanation of why this method was chosen to make it clear why the direct comparison was not used. Response: Thank the reviewer’s comment. Yes, we have initially contemplated using this intuitive approach that directly comparing corresponding intervals between years (i.e. bi-week 8 of 2020 compared to bi-week 8 of 2019). But on a second thought, as Reviewer #2 raised concern in “Question 2” that bi-week scale inflates the number of regression models required and dilutes the sample size being examined in each of these statistical models, leading to potentially erroneous inferences (i.e. the multiple comparisons problem) and imprecise odds ratio estimates, this intuitive approach that directly comparing corresponding intervals then was not used. Aside from indirect comparison in Table 2, we also used Figure 3B to compare between groups whether the percentage of uncomplicated & complicated appendicitis in patients with appendicitis increased during the COVID-19 pandemic. Figure 3A depicts the raw data for comparison.

4. The language to describe severity of appendicitis is not used consistently throughout. The severity categories are “normal, mild, uncomplicated and complicated” in some places (e.g. the Dependent variables section), and “normal, mild, moderate or severe” in others (e.g. Abstract and Results section). Clarification of these categories will improve readability. I strongly suggest clarifying these terms and explicitly defining the outcome of interest for the logistic regression analysis in the Dependent variable section. Response: Yes, we thank the reviewer’s comment. We have made changes in the revised manuscript. “The dependent variable was uncomplicated and complicated appendicitis. The severity of appendicitis was classified as normal, mild, uncomplicated (moderate), and or complicated (severe) appendicitis [14].” (2nd sentence, 3rd paragraph, Materials and methods section) “The Ppercentage of moderate/severe uncomplicated and complicated appendicitis (excluding mild appendicitis or normal appendix) in the study group increased between February 26 and March 2410, as well as between April 228 and May 5 April 21.” (last 2nd sentence, Results of Abstract). “The Nnumber of patients with moderate/severe uncomplicated and complicated appendicitis would not is unlikely be likely to decrease due to the COVID-19 outburst outbreak.” (3rd sentence, 2nd paragraph, Introduction section). “Compared to with the control group, the percentage difference of severe uncomplicated and complicated appendicitis (excluding mild appendicitis/normal appendix) for in the study group (vs. the control group) increased up to by more than 40% during the fifth bi-week 5 (February 26–March 10) and eighth bi-week 8 (April 8–April 21) (Figure 3).” (last sentence, 2nd paragraph, Results section).

5. I suggest avoiding using the word “significant” when reporting differences, as there is evidence that the p-value significance criteria of <0.05 is too high and contributing to the reproducibility crisis in research (see Benjamin, D.J., Berger, J.O., Johannesson, M. et al. Redefine statistical significance. Nat Hum Behav 2, 6–10 (2018). Response: Yes, we thank the reviewer’s comment and reference relating statistical significance. We have made a change in the revised manuscript. “However, a such finding of significant increase increasing of in OR for the severity of appendicitis was shown not observed in neither the univariate nor or multivariate analysis analyses of the control group.” (last 2nd sentence, last paragraph, Results section). 

6. “Multivariate logistic regression analysis was conducted to assess the possibility of getting ruptured (or impending) ruptured appendicitis at different time period”. This does not clearly define the outcome of interest using the established appendicitis severity ratings, and does not describe the groups being compared, and needs to be revised for clarity Response: Yes, we thank the reviewer’s comment. We have made a modification in the revised manuscript. “Univariate and Mmultivariate logistic regression analysies was were conducted to assess the possibility of getting ruptured (or impending) ruptured uncomplicated and complicated appendicitis (vs. normal & mild appendicitis) at different time periods. The dependent variable was uncomplicated & complicated appendicitis.” (6th, 4th paragraph, Materials and methods section).

7. “Severity of acute appendicitis might increase during COVID-19 pandemic, probably due to decrease attitude to seek help in patient with mild appendicitis (or abdominal pain)”. The wording is very strong and risks overstating findings. The use of “probably” should be substituted with more cautious wording, such as “potentially” or “may be”. Response: Yes, we thank the reviewer’s comment. We therefore have made a change in the revised manuscript after obtaining assistance in English writing. “The Sseverity of acute appendicitis might increase during the COVID-19 pandemic, probably because due to decrease attitude to seek help in patients with mild appendicitis (or abdominal pain) may hesitate to seek help.” (Conclusion, Abstract).

8. I would recommend the authors include a limitations section in their Discussion section to highlight any caveats on their conclusions. Response: Yes, we thank the reviewer’s comment. We have added a limitation in the revised manuscript. 

9. Table 1 suggestions:

- Include subheadings to indicate which characteristics relate to patients, treatments and hospitalization.

- Fecolith should have a “No” category.

- Consider reporting median and interquartile range for some continuous outcomes such as cost and length of hospital stay. This is to be consistent with the Results section (“Median hospital stay in the study group was 3 (1–16) days, vs. 3 (1–28) days in the control group (p = 0.818)”) and because these variables are often non-normally distributed, so the median and interquartile range better capture the spread of the variable.

-In the Statistical analysis section, include detail about method used to get the p-value comparing continuous outcomes in Table 1

Response: Yes, we thank the reviewer’s comment. A. We have made a change regarding subheadings (” Patient characteristics”,” Treatment characteristics”,” Hospitalization characteristics”) in Table 1 of the revised manuscript. B. We have added “No” category and their corresponding number in Table 1 of the revised manuscript. C. We have added median and interquartile range of cost and length of hospital stay in the revised Table 1. D. We have made a modification in the revised manuscript. “The numerical variables (medical expenditure and hospital stay) were compared using Student’s t-test.” (4th sentence, 4th paragraph, Materials and methods section).

10. Table 2 caption should include details about univariable analysis and what the comparator groups are. Response: Yes, we thank the reviewer’s comment. We have made a change according to reviewer’s suggestion in the revised manuscript. “Table 2 Univariate and Mmultivariate logistic regression analyses of odds ratios for uncomplicated/ and complicated appendicitis (vs. normal/ and mild appendicitis)” (Caption, Table 2).

11. Table 3 needs an appropriate caption to indicate stratification by surgery type. Response: Yes, we thank the reviewer’s comment. We have made a change according to reviewer’s suggestion in the revised manuscript. “Table 3 Baseline characteristics Medical expenditure and hospital stay of patients undergoing appendectomy according to surgery type” (Caption, Table 3).

12. For Figure 1, the relevance of world COVID-19 case and fatality figures is not clear. Do the authors anticipate rising cases around the world would have some impact on hospital attendance in Taiwan in the absence of local cases? E.g. Was there ongoing concern about possible imported cases? Response: Yes, we thank the reviewer’s comment. Figure 1 combines the COVID-19 cases and fatality figures between the world and Taiwan. This figure illustrates that the surges in Taiwan preceded the worldwide pandemic a little bit earlier but became indolent thereafter. So it probably hints that local cases of COVID-19 would have some impact on hospital attendance in Taiwan initially, but not in the last half period of time. Compared with the world in the same six months, this figure showed that the confirmed cases of COVID-19 in Taiwan did not increase in parallel with the worlds in all the six months except for one or two surges in one month. That’s why we chose a “bi-week” scale rather than a “month” scale in the timeframe. The reason why this relationship between Taiwan and the world illustrated in Figure 1 is probably due to countermeasures taken to prevent disease dissemination in Taiwan in the very beginning of the pandemic which was beyond the scope our study purpose. 

13. For Figure 2, I suggest that the x-axis include dates ranges instead of bi-week number for easier comparison with the case numbers provided by date in Figure 1. The reason for including the “reference” line (July 2019-December 2020) is not clear, as this time period was not discussed in the manuscript at all – could the authors please justify this? Response: Yes, we thank the reviewer’s comment. For figure 2, we have added dates ranges in the revised Figure 2, and removed the “reference line (July 2019-December 2019)” from Figure 2 according to reviewer’s recommendation. 

14. Figure 3 suggestions:

- A. The use of percentage differences should be replaced with actual raw value differences in B, as percentages can be misleadingly large when there are few appendicitis cases.

- B. As with Figure 2, suggest labelling each bi-week with the corresponding date range for easier comparison with the case numbers shown in Figure 1, as well as discussion of why the July 2019-December 2020 timeframe was included. Response: Yes, we thank the reviewer’s kind comment. A. In Figure 3, 3A shows the actual raw value of mild (& normal), uncomplicated, and complicated appendicitis between two groups. Figure 3B supplements the information from Figure 3A by percentage difference between two groups. Figure 3B could be created by percentage difference of mild appendicitis or uncomplicated and complicated appendicitis, which we chose the latter to illustrate. Although we found percentage of mild appendicitis decreased (ie. the percentage of uncomplicated and complicated appendicitis increased), we finally chose the figure of uncomplicated and complicated appendicitis as Figure 3B since it shows two surges which somewhat resemble Figure 1A. Therefore, we would like to present percentage difference as original Figure 3B shows. B. For figure 3, we have added dates ranges in the revised Figure 3, and removed the data of cases between July 2, 2019 and December 30, 2019 in the revised Figure 3A according to reviewer’s recommendation to avoid misinterpretation.

---

## [Decision Letter · Decision Letter 1]

5 Nov 2021

PONE-D-21-07736R1

Did the severity of appendicitis increase during the COVID-19 pandemic?

PLOS ONE

Dear Dr. Chang,

Thank you for submitting your manuscript to PLOS ONE. After careful consideration, we feel that it has merit but does not fully meet PLOS ONE’s publication criteria as it currently stands. Therefore, we invite you to submit a revised version of the manuscript that addresses the points raised during the review process.

 The quality of your article have been greatly improved. However, our reviewer still raised some concerns. We look for your corresponding response and revisions. 

We look forward to receiving your revised manuscript.

Kind regards,

Chun Chieh Yeh, M.D., Ph.D.

Academic Editor

PLOS ONE

Journal Requirements:

Additional Editor Comments (if provided):

Thanks for your revisions. The quality of your article have been greatly improved. However, our reviewer still raised some concerns. We look for your corresponding response and revisions.

Reviewers' comments:

Reviewer's Responses to Questions

**Comments to the Author**

1. If the authors have adequately addressed your comments raised in a previous round of review and you feel that this manuscript is now acceptable for publication, you may indicate that here to bypass the “Comments to the Author” section, enter your conflict of interest statement in the “Confidential to Editor” section, and submit your "Accept" recommendation.

Reviewer #1: All comments have been addressed

Reviewer #2: (No Response)

2. Is the manuscript technically sound, and do the data support the conclusions?

Reviewer #1: Yes

Reviewer #2: Partly

3. Has the statistical analysis been performed appropriately and rigorously? 

Reviewer #1: Yes

Reviewer #2: No

4. Have the authors made all data underlying the findings in their manuscript fully available?

Reviewer #1: Yes

Reviewer #2: No

5. Is the manuscript presented in an intelligible fashion and written in standard English?

Reviewer #1: Yes

Reviewer #2: Yes

6. Review Comments to the Author

Reviewer #1: Comments to the Author PONE-D-21-04326R1

The authors have completely addressed the comments, and added response to text and study limitation.

However, reference 4 and reference 10 are repeated. Need revision.

Reviewer #2: I thank the editors and authors for the chance to review the revised manuscript, “Did the severity of appendicitis increase during the COVID-19 pandemic?”. The quality of the paper has improved significantly with revision. The authors have addressed many of the concerns from the initial review, however I have some additional concerns on the revised manuscript. There is one remaining major comment, and several minor comments:

Major comment

The authors have maintained the approach dividing the data into bi-week periods for comparing the number of moderate or severe appendicitis presentations within the study and control groups. As mentioned in the previous review, there are serious issues with this method from a statistical perspective due to the large number of regression models required. Although the authors do mention these limitations in the Discussion, there should also be a clear statement in the Methods that no adjustment was made for multiple comparisons. I would also recommend an explicit mention of the small sample size relative to the number of parameters included in the model, leading to imprecise estimates for some variables. Additionally, I believe the paper would benefit greatly by providing a comparison of the overall difference in moderate or severe appendicitis over 6 months between the study groups. This is a simple, less underpowered analysis which is important to provide context to the readers about the “overall” effect of the pandemic and to frame the results provided in the bi-week approach.

Minor comments

1. Abstract: “The mean age was 46.2 + 19.8 years; male predominance was observed was observed (52.1%, 160/307).”

‘Was observed’ is repeated twice. ‘Predominance’ is not the right word in this sentence as there are only slightly more males than females.

2. Materials and Methods, Dependent Variable: “The dependent variable was uncomplicated and complicated appendicitis.”

Suggest moving this sentence to after explanation of the different levels of appendicitis, and refining so it’s clear that this is a binary variable e.g. “The dependent variable was a binary indicator of uncomplicated and complicated appendicitis (0 – normal, mild; 1 – uncomplicated, complicated).”

3. Materials and Methods, Statistical Analysis: “Patient characteristics of (excluding age), treatment (excluding surgery time), and disease variables aree reported as numbers and percentages.”

There are typos in this sentence – ‘of’ should be removed before “(excluding age)” and ‘aree’ should be replaced with ‘are’.

4. Results: “Compared with the control group, the number of surgeries for appendicitis was relatively low between January 29 and April 21 in the study group (Figure 2).”

I would disagree with this statement, in my reading I see the numbers as approximately equivalent for the 25th of March to the 21st of April. Therefore, to avoid this subjectivity in the results section, I suggest reporting the actual number of surgeries in this period for either group rather than just referring to the figure.

5. Results: “Surgery was performed under general anesthesia in approximately 80% of the patients.”

Should replace ‘approximate’ with ‘approximately’.

6. Results and Table 1: “The mean operation time was 67 + 27 min (range: 20–167 min). Dirty or contaminated wounds were observed in a quarter (25.1%, 77/307)”

The upper limit of the operation time is 168 minutes in Table 1 – please fix this so the values match. Table 1 also has a value of 7/307 for dirty or contaminated wounds, which looks like a typo.

7. Results: “Compared with the control group, the percentage difference of uncomplicated and complicated appendicitis in the study group (vs. the control group) increased by more than 40% during the fifth bi-week (February 26–March 10) and eighth bi-week.”

Yes, but raw numbers of presentations must also be reported in this sentence to provide context to the percentage change. In both bi-week 5 and 8, the percentage change seems to be primarily driven by having fewer mild/normal cases rather than an increase in uncomplicated/complicated appendicitis. Providing the raw numbers in addition to percentage difference is necessary for transparency in the text as it was for Figure 3.

8. Results and Table 2: “April 8–April 21 (OR = 11.25, 95% CI: 1.17–>100 , P = 0.036)”

Please report the upper limit of the 95% CI as a value, not ‘>100’. This also applies to bi-week 5 in the study group multivariate analysis.

9. Results: “However, a such increase in OR for the severity of appendicitis was not observed in the univariate or multivariate analyses of the control group.”

This is not accurate for bi-week 2 as the estimated OR in the control group is 2.81 (univariate) and 6.38 (multivariate) which is quite similar to the ORs estimated in the study group (3.25 (univariate) and 5.81 (multivariate)). I believe the authors are discussing the 95% CI in this case, but the comment implies that the magnitude of the effect is small for the control group and so is inappropriate for this specific comparison.

10. Discussion and Conclusion:

I believe the authors need to further review/soften some of the language around the findings from their study considering the limitations of the statistical methods used. For example:

- “This occurred when the number of confirmed cases of COVID-19 surged.”

- “The severity of appendicitis also increased, which inversely paralleled the number of surgeries for appendicitis during the pandemic.”

- “Therefore, during the COVID-19 pandemic, the number of surgeries for mild appendicitis has decreased, and the severity of appendicitis has increased.”

- “The number of patients with acute appendicitis seeking surgical treatment decreased as confirmed cases of COVID-19 surged.”

These kind of conclusions overstate the findings as a number of the ORs during the pandemic period in the study group have 95% CIs which are less than 1 (e.g. bi-week 6, 7 and 9). I recommend these sentences are amended to indicate the uncertainty evidenced in the 95% CIs.

7. PLOS authors have the option to publish the peer review history of their article (what does this mean?). If published, this will include your full peer review and any attached files.

Reviewer #1: No

Reviewer #2: No

---

## [Author Response · Author response to Decision Letter 1]

23 Nov 2021

Author’s point-by-point response to review comments (PONE-D-21-07736R1) : 

We have carefully revised the manuscript in view of the editorial and reviewers’ comments as outlined in detail below.

Reviewer’s comments:

Reviewer #1: The authors have completely addressed the comments, and added response to text and study limitation.

1. However, reference 4 and reference 10 are repeated. Need revision. Response: Thanks to reviewer’s comment. We have updated references (by deleting repeated reference 10 and updating numbering of references) in the resubmitted manuscript.

Reviewer #2: The quality of the paper has improved significantly with revision. The authors have addressed many of the concerns from the initial review, however I have some additional concerns on the revised manuscript.

Major comment

1. As mentioned in the previous review, there are serious issues with this method from a statistical perspective due to the large number of regression models required. Although the authors do mention these limitations in the Discussion, there should also be a clear statement in the Methods that no adjustment was made for multiple comparisons. I would also recommend an explicit mention of the small sample size relative to the number of parameters included in the model, leading to imprecise estimates for some variables. Additionally, I believe the paper would benefit greatly by providing a comparison of the overall difference in moderate or severe appendicitis over 6 months between the study groups. This is a simple, less underpowered analysis which is important to provide context to the readers about the “overall” effect of the pandemic and to frame the results provided in the bi-week approach.

Response: Yes, we thank the reviewer’s informative comment. 

A. We have added a statement regarding no adjustment was made for multiple comparisons in the Methods resection of the revised manuscript. “No adjustment was made for multiple comparisons when the sample size relative to the number of parameters was small.” (last 3rd sentence, last paragraph, Materials and methods section) 

B. We have added a statement regarding mention of the small sample size relative to the number of parameters in the Methods resection of the revised manuscript. “No adjustment was made for multiple comparisons when the sample size relative to the number of parameters was small.” (last 3rd sentence, last paragraph, Materials and methods section)

C. We have also added comparison of the overall difference in moderate or severe appendicitis over 6 months between the study groups in the Results section of the revised manuscript. “Although the percentage of uncomplicated or complicated appendicitis was lower higher in the study group during the COVID-19 pandemic than that of the control group (68.5% vs. 58.9%), the difference was insignificant (P = 0.213). For comparison, the percentage of uncomplicated or complicated appendicitis over 6 months’ period between the study and control groups was 66.9 % (103/163).” (2nd and third sentences, 2nd paragraph, Results section).

Minor comments

1. Abstract: “The mean age was 46.2 + 19.8 years; male predominance was observed was observed (52.1%, 160/307).”

‘Was observed’ is repeated twice. ‘Predominance’ is not the right word in this sentence as there are only slightly more males than females.

Response: Yes, we appreciate the reviewer’s comment. We have delete “; male predominance was observed was observed (52.1%, 160/307)” in the revised manuscript. 

2. Materials and Methods, Dependent Variable: “The dependent variable was uncomplicated and complicated appendicitis.”

Suggest moving this sentence to after explanation of the different levels of appendicitis, and refining so it’s clear that this is a binary variable e.g. “The dependent variable was a binary indicator of uncomplicated and complicated appendicitis (0 – normal, mild; 1 – uncomplicated, complicated).”

Response: Yes, we thank the reviewer’s comment. We have deleted the 1st sentence (dependent variable paragraph, Materials and methods section) and added a sentence in the revised manuscript. “The dependent variable was a binary indicator of uncomplicated and complicated appendicitis (0 – normal, mild; 1 – uncomplicated, complicated).” (5th sentence, dependent variable paragraph, Materials and methods section)

3. Materials and Methods, Statistical Analysis: “Patient characteristics of (excluding age), treatment (excluding surgery time), and disease variables aree reported as numbers and percentages.”

There are typos in this sentence – ‘of’ should be removed before “(excluding age)” and ‘aree’ should be replaced with ‘are’.

Response: Yes, we thank the reviewer’s comment. We have corrected these typos in the revised manuscript.

4. Results: “Compared with the control group, the number of surgeries for appendicitis was relatively low between January 29 and April 21 in the study group (Figure 2).”

I would disagree with this statement, in my reading I see the numbers as approximately equivalent for the 25th of March to the 21st of April. Therefore, to avoid this subjectivity in the results section, I suggest reporting the actual number of surgeries in this period for either group rather than just referring to the figure.

Response: Yes, we thank the reviewer’s comment. We have added the actual number of surgeries in this period for either group in the revised manuscript. “Compared with the control group (70/158, 44.3%), the number of surgeries for appendicitis was relatively low between January 29 and April 21 (49/149, 32.9%) in the study group (Figure 2).” (6th sentence, 1st paragraph, Results section)

5. Results: “Surgery was performed under general anesthesia in approximately 80% of the patients.”

Should replace ‘approximate’ with ‘approximately’.

Response: We appreciate the reviewer’s comment. We have made a correction in the revised manuscript. 

6. Results and Table 1: “The mean operation time was 67 + 27 min (range: 20–167 min). Dirty or contaminated wounds were observed in a quarter (25.1%, 77/307)”

The upper limit of the operation time is 168 minutes in Table 1 – please fix this so the values match. Table 1 also has a value of 7/307 for dirty or contaminated wounds, which looks like a typo.

Response: Yes, we appreciate the reviewer’s comment. We have made corrections for these typos in the revised manuscript.

7. Results: “Compared with the control group, the percentage difference of uncomplicated and complicated appendicitis in the study group (vs. the control group) increased by more than 40% during the fifth bi-week (February 26–March 10) and eighth bi-week.”

Yes, but raw numbers of presentations must also be reported in this sentence to provide context to the percentage change. In both bi-week 5 and 8, the percentage change seems to be primarily driven by having fewer mild/normal cases rather than an increase in uncomplicated/complicated appendicitis. Providing the raw numbers in addition to percentage difference is necessary for transparency in the text as it was for Figure 3.

Response: Yes, we thank the reviewer’s comment. We have provided the raw numbers in addition to percentage difference in the revised manuscript. “Compared with the control group, the percentage difference of uncomplicated and complicated appendicitis in the study group (88.9%, 8/9 vs. the control group 42.9%, 6/14) increased by more than 40% during the fifth bi-week (February 26–March 10) (88.9%, 8/9 vs. the control group 42.9%, 6/14) and eighth bi-week (April 8–April 21) (90.0%, 9/10 vs. the control group 46.7%, 7/15) (Figure 3)” (last sentence, 2nd paragraph, Results section)

8. Results and Table 2: “April 8–April 21 (OR = 11.25, 95% CI: 1.17–>100 , P = 0.036)”

Please report the upper limit of the 95% CI as a value, not ‘>100’. This also applies to bi-week 5 in the study group multivariate analysis.

Response: Yes, we thank the reviewer’s comment. We have reported these values in the Results and Table 2 of the revised manuscript.

9. Results: “However, a such increase in OR for the severity of appendicitis was not observed in the univariate or multivariate analyses of the control group.”

This is not accurate for bi-week 2 as the estimated OR in the control group is 2.81 (univariate) and 6.38 (multivariate) which is quite similar to the ORs estimated in the study group (3.25 (univariate) and 5.81 (multivariate)). I believe the authors are discussing the 95% CI in this case, but the comment implies that the magnitude of the effect is small for the control group and so is inappropriate for this specific comparison.

Response: Yes, we appreciate the reviewer’s comment. From Table 2, we never found the ORs (for the uncomplicated/complicated appendicitis) over 7 in the right two columns of the control group (neither univariate nor multivariate), which P values all insignificant. However, we noted that many ORs (for the uncomplicated/complicated appendicitis) overpass 7 in the left two columns of the study group (univariate & multivariate) and many of them were significant. Still, we have rephrased this sentence to minimize misunderstanding in the revised manuscript. “However, such a significant increase in OR (bi-weekly) for the severity of appendicitis was not observed in the univariate or multivariate analyses of the control group.” (last 2nd sentence, last paragraph, Results section)

10. Discussion and Conclusion:

I believe the authors need to further review/soften some of the language around the findings from their study considering the limitations of the statistical methods used. For example:

- “This occurred when the number of confirmed cases of COVID-19 surged.”

- “The severity of appendicitis also increased, which inversely paralleled the number of surgeries for appendicitis during the pandemic.”

- “Therefore, during the COVID-19 pandemic, the number of surgeries for mild appendicitis has decreased, and the severity of appendicitis has increased.”

- “The number of patients with acute appendicitis seeking surgical treatment decreased as confirmed cases of COVID-19 surged.”

These kind of conclusions overstate the findings as a number of the ORs during the pandemic period in the study group have 95% CIs which are less than 1 (e.g. bi-week 6, 7 and 9). I recommend these sentences are amended to indicate the uncertainty evidenced in the 95% CIs.

Response: Yes, we thank the reviewer’s comment. We have modified these sentences in the revised manuscript. 

“This probably occurred when the number of confirmed cases of COVID-19 surged in Taiwan. In addition, the severity of appendicitis also increased, which appeared to inversely paralleled the number of surgeries for appendicitis during the pandemic.” (4th and 5th sentences, 1st paragraph, Discussion section) 

“Therefore, during the COVID-19 pandemic, the number of surgeries for mild appendicitis has seemed to decreased, and the severity of appendicitis has increased.” (last sentence, 1st paragraph, Discussion section)

“The number of patients with acute appendicitis seeking surgical treatment might decreased as confirmed cases of COVID-19 surged,…….”(2nd sentence, Conclusions section)

Response to editors:

We also have updated references and reference type in the revised manuscript to conform the requirement of the journal.

---

## [Decision Letter · Decision Letter 2]

28 Jan 2022

Did the severity of appendicitis increase during the COVID-19 pandemic?

PONE-D-21-07736R2

Dear Dr. Chang,

We’re pleased to inform you that your manuscript has been judged scientifically suitable for publication and will be formally accepted for publication once it meets all outstanding technical requirements.

Kind regards,

Chun Chieh Yeh, M.D., Ph.D.

Academic Editor

PLOS ONE

Additional Editor Comments (optional):

Thanks for your specific responses to our referees' comments and we are glad to accept this article for publication.

However, only one minor revision was suggested and we wish you could revise it at the final draft.

Our referee had one very minor remaining comment on one sentence:

Results: “The median hospital stay in the study group was 3 (1–16) days and in the control group 3 (1–28) days (P = 0.818).”

Please amend to indicate that the value in brackets is the range.

Reviewers' comments:

Reviewer's Responses to Questions

**Comments to the Author**

1. If the authors have adequately addressed your comments raised in a previous round of review and you feel that this manuscript is now acceptable for publication, you may indicate that here to bypass the “Comments to the Author” section, enter your conflict of interest statement in the “Confidential to Editor” section, and submit your "Accept" recommendation.

Reviewer #1: All comments have been addressed

Reviewer #2: All comments have been addressed

2. Is the manuscript technically sound, and do the data support the conclusions?

Reviewer #1: Yes

Reviewer #2: Yes

3. Has the statistical analysis been performed appropriately and rigorously? 

Reviewer #1: Yes

Reviewer #2: Yes

4. Have the authors made all data underlying the findings in their manuscript fully available?

Reviewer #1: Yes

Reviewer #2: Yes

5. Is the manuscript presented in an intelligible fashion and written in standard English?

Reviewer #1: Yes

Reviewer #2: Yes

6. Review Comments to the Author

Reviewer #1: The authors have completely addressed the comments and added responses to text and study limitation.

The references are updated.

Reviewer #2: All comments from the previous review have been addressed by the authors and I would recommend accepting the manuscript for publication. I have one very minor remaining comment on one sentence:

Results: “The median hospital stay in the study group was 3 (1–16) days and in the control group 3 (1–28) days (P = 0.818).”

Please amend to indicate that the value in brackets is the range.

7. PLOS authors have the option to publish the peer review history of their article (what does this mean?). If published, this will include your full peer review and any attached files.

Reviewer #1: No

Reviewer #2: No

---

## [Editor Report · Acceptance letter]

2 Feb 2022

PONE-D-21-07736R2 

Did the severity of appendicitis increase during the COVID-19 pandemic? 

Dear Dr. Chang:

I'm pleased to inform you that your manuscript has been deemed suitable for publication in PLOS ONE. Congratulations! Your manuscript is now with our production department. 

Kind regards, 

on behalf of

Dr. Chun Chieh Yeh 

Academic Editor

PLOS ONE